# Investigating Physiological Responses During Driving as Potential Biomarkers of Cognitive Decline in Seniors Using Decision Tree Ensemble Modeling

Anu M. Kumar
*Dept. of Electrical and Computer Engineering*
*University of Michigan*
MI, USA
kumaram@umich.edu

Yi Lu Murphey
*Dept. of Electrical and Computer Engineering*
*University of Michigan*
MI, USA
yilu@umich.edu

Amanda Cook Maher
*Dept. of Psychiatry*
*University of Michigan Medical School*
MI, USA
amhco@med.umich.edu

Carol Persad
*Dept. of Psychiatry*
*University of Michigan Medical School*
MI, USA
cpersad@med.umich.edu

Robert Koeppe
*Dept. of Radiology*
*University of Michigan Medical School*
MI, USA
koeppe@med.umich.edu

Bruno Giordani
*Dept. of Psychiatry*
*University of Michigan Medical School*
MI, USA
giordani@med.umich.edu

*Abstract*—Early identification of individuals at risk for Alzheimer's disease is essential to improve treatment effectiveness. Cerebrospinal fluid analyses and positron emission tomography (PET) scans are commonly used to detect the presence of beta-amyloid and tau, which are associated with an increased risk of conversion to Alzheimer's disease. However, these biomarker tests are expensive and involve invasive procedures. Researchers are working towards discovering easily measurable biomarkers to detect individuals at risk, but only a few have been identified thus far. There is a need to discover biomarkers that are cost-efficient and non-invasive to test. We propose a machine learning approach for discovering potential risk biomarkers of Alzheimer's disease through the analysis of physiological responses to the cognitively complex task of driving by using decision tree ensemble techniques. Though driving patterns in early Alzheimer's have been previously studied, physiological responses of cognitively normal seniors during driving remain unexplored. As a first step, we measure heart rate, electrodermal activity, and temperature responses to several driving events, such as right turns and roundabouts, of seniors with and without elevated PET beta-amyloid levels to explore the relationship between these physiological responses and amyloid level. Data were collected from 26 participants with elevated beta-amyloid and 28 without. We used four machine learning algorithms for classification: Random Forest, Extra Trees, AdaBoost, and XGBoost, and developed a novel methodology to extract significant features from these models. By doing so, we successfully identified five risk biomarkers most influential in differentiating the two groups with and without elevated beta-amyloid.

*Index Terms*—Beta-amyloid, Decision Tree Ensemble, Driving and Alzheimer's, Physiological Signals, Risk Biomarker Discovery

## I. INTRODUCTION

As the population of senior citizens in the US continues to grow, the number of Americans with dementia is also on the rise. It is estimated that in 2050, the population of Americans age 65 and older will increase from 58 million to 82 million in 2022 [1]. About 6.9 million senior citizens are living in the US with Alzheimer's disease (AD) in 2024, and this number is predicted to increase to 12.7 million by 2050 [1]. AD is the most common cause of dementia, accounting for two-thirds of all dementia diagnoses [3], and results in the loss of cognitive functioning [2], which hinders the ability to perform daily life activities.

AD is a neurodegenerative disease with distinct pathological characteristics. These include accumulations of beta-amyloid ($A\beta$) plaques outside neurons [1], which are caused by the breakdown of the amyloid precursor protein [5], and abnormal accumulations of the tau protein inside neurons, which are known as neurofibrillary tangles. The diagnosis of AD is a complex, multifactorial process that includes the assessment of key biomarkers such as neurodegeneration, elevated $A\beta$, and tau. Definite tests for these biomarkers include cerebrospinal fluid (CSF) analyses and PET scans [6, 9]. However, these diagnostic methods can be quite expensive, require specialized equipment, and often involve invasive procedures for the individual. Exploration of novel risk biomarkers that are reliable, non-invasive, and cost-efficient are necessary to identify individuals at higher risk of developing AD in the future.

Many studies have discussed the association between elevated $A\beta$ levels, as identified through PET imaging, and an increased risk of conversion to mild cognitive impairment (MCI) due to AD [21, 27]. However, $A\beta$ levels may increase 15 to 20 years or more before symptoms become apparent, coinciding with increases in tau levels. Cognitively normal seniors with elevated $A\beta$ may exhibit subtle yet quantifiable changes in their daily activities. Looking for changes in more complex activities of daily living may provide insights into one's chances of developing cognitive decline in the future. Specifically, driving is an incredibly complex and cognitively demanding task that is regularly undertaken by many older adults and may therefore be an ideal daily activity for monitoring early decline. The early impact of AD on driving has been studied by researchers in the past. As an example, Wadley et al. [7] compared the driving

This work was supported in part by the NIH/NIA Grant R01AG068338 (Giordani, Murphey, Persad, MPIs) and P30AG072931 (Paulson, PI).

performance of individuals with and without MCI and found that drivers with MCI had a greater likelihood of obtaining suboptimal ratings for lane control and left turns. Similarly, Stinchcombe et al. [8] analyzed the simulated driving performance at intersections for drivers with mild AD and concluded that they commit most of their errors during the approach to the intersection. These studies focused on analyzing the driving behavior of individuals with an early MCI or AD diagnosis who have started to display symptoms. The physiological signals of cognitively normal individuals with elevated $A\beta$ burden remain unexplored. For example, increasing heart rate or electrodermal activity may reflect anxiousness or increased situational awareness in key driving situations. Given the link between PET detected $A\beta$ levels and the risk of developing AD, comparing the physiological responses during driving of seniors with and without elevated $A\beta$ may reveal distinctive characteristics of drivers with elevated $A\beta$. Identifying cost-effective alternatives for the early detection of cognitive decline would greatly enhance the accessibility of early AD diagnosis. Consequently, we chose to study the physiological signals of seniors with and without increased $A\beta$ during driving to discover potential novel susceptibility biomarkers of cognitive decline due to AD. In this study, we explore whether senior drivers with higher $A\beta$ levels exhibit different physiological responses to key driving scenarios with increased cognitive demand (e.g., freeway entrances, roundabouts) compared to senior drivers with no elevation in $A\beta$.

Artificial intelligence (AI) and machine learning (ML) methods have shown promising results for biomarker identification [15]. Specifically, decision trees, a form of supervised learning in AI and ML, have been widely used for biomarker discovery and classification by many studies [10, 15]. As an example, Hamsagayathri et al. [11] used J48 and Classification and Regression Trees (CART) decision tree algorithms to classify breast cancer. An alternating decision tree with principal component analysis was deployed for early heart disease classification by Jabbar et al. [12], whereas Li et al. [13] utilized the Extreme Gradient Boosting tree (XGBoost), which is a form of gradient boosting decision trees, to identify prognostic biomarkers of cancer. Over the years, many studies have unveiled the ability of decision trees and decision tree ensembles to provide reliable biomarker identification and classification results.

The aim of this study is to analyze the physiological responses of senior drivers with normal cognition with and without elevated $A\beta$ to discover potential risk biomarkers of impending cognitive decline using multiple ML ensembles of decision trees (EoDT) techniques. We utilize the four following EoDT algorithms: Random Forest, Extra Trees, AdaBoost, and XGBoost, and compare each method's accuracy of classifying seniors with elevated $A\beta$ and ranking of feature importances to identify the best risk biomarkers of cognitive decline. A risk or susceptibility biomarker is defined as a biomarker used to identify an individual, currently asymptomatic, at risk of developing a given disease or condition in the future [4].

The remainder of the paper is organized as follows. Related work is discussed in Section II. Data collection and processing methods are described in Section III. The EoDT models used, experimental methods, and biomarker selection procedure are explained in Section IV. Experimental results and discovered potential risk biomarkers are presented in Section V. Lastly, the future scope and final remarks are discussed in section VI.

## II. RELATED WORK

In this section, we discuss related work on extracting and analyzing physiological features to identify early cognitive decline and AD.

In order to differentiate between seniors with and without AD, Vicchietti et al. [16] utilized computational electroencephalographic (EEG) signal analysis. The EEG database was obtained from Florida State University and pre-processed using the Wavelet Transform. The researchers applied six different computational time-series analysis methods, including wavelet coherence and fractal dimension, to the EEG data for feature extraction and compared the robustness of each method in identifying seniors with AD. Each method was evaluated using area under the curve and the ANOVA p-value to assess its ability to detect AD. It was concluded that the Quantiles Graph method produced the highest accuracy, sensitivity, and specificity scores.

The study conducted by Dieffenderfer et al. [14] developed a wearable system that longitudinally monitors physiological and behavioral signals, such as heart rate variability and electrodermal activity (EDA). This system aims to enable the early detection of AD and related dementias. Here, cognitive stress was measured using EDA, skin temperature, and photoplethysmography. The researchers designed a wrist band to collect these signals and a waist patch to collect gait and speech data. The researchers in this study plan to perform ML-based analyses on the data collected using this system as future work, in order to detect early AD and related dementias.

The extraction and analysis of physiological signals using ML and other approaches have significant potential for identifying individuals at risk of developing cognitive decline. While physiological signals have been studied under resting and longitudinal conditions, they have not been evaluated during the cognitively demanding task of driving. Despite extensive research on identifying early cognitive decline, most studies have only focused on individuals after the onset of symptoms. To address this gap, we have studied the physiological signals of cognitively normal $A\beta$ positive and negative seniors during key driving events using decision tree ensembles to identify risk biomarkers of cognitive decline.

## III. DATA COLLECTION AND PROCESSING

### A. Participants

Fixed course driving trips from 54 participants were collected for this study. All participants were diagnosed as cognitively normal via consensus conference based on neuropsychological testing, informant report, and medical and neurological evaluations by a clinician. Each participant underwent PiB-PET scanning and the level of $A\beta$ in their brain was quantified based on the centiloid scale: centiloid values of 10 or lower

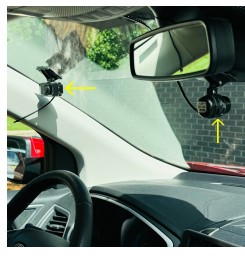

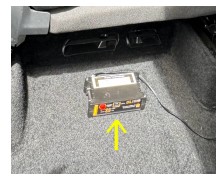

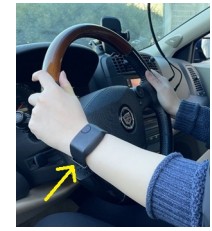

(a) Video Cameras

(b) Race Technology DL1 CLUB Data Logger

(c) Empatica E4 Wristband

Fig. 1: Devices used for Data Acquisition

are indicative of no elevation in $A\beta$ ($A\beta$ negative), while centiloid values of 20 or higher indicate elevated $A\beta$ burden ($A\beta$ positive). One group consisted of 26 participants with elevated $A\beta$ levels. The mean age of this group was 73 years (standard deviation (SD)=4.16) with a range of 66 to 81 years, and consisted of 15 men and 11 women. The other group consisted of 28 participants with no elevation in $A\beta$. The mean age of this group was 73 years (SD=4.98) with a range of 65 to 85 years, and included 10 men and 18 women. Every participant held a valid driver's license.

### B. Fixed Course Drive

All drivers completed a fixed course drive during the daytime. They followed a suburban driving route in their own vehicles, accompanied by an evaluator. The fixed course route was 7.1 miles long and consisted of seven left turns, three right turns, three intersections with traffic signals, six intersections with stop signs, three yield signs, one freeway entrance ramp, one freeway exit ramp, three roundabouts, three intersections with signs, five intersections without signs, and two parking lots.

The fixed course drive, encompassing a consistent set of roadway types and conditions, provided a standardized evaluation to ensure fair comparison among all drivers. Evaluating driving behavior through naturalistic drives may pose the issue of fairness, for example, one driver's route may consist of a route without any freeways and minimal intersections, while another driver's route may consist of multiple busy intersections. In this study, we kept the route consistent for all participants in order to ensure the reliability of our findings.

### C. Data Collection

Three types of data were collected from each participant during their fixed course drive: video, vehicular, and physiological data. Video data were collected using two video cameras mounted inside the driver's own car, one facing the driver to record the driver's upper body and face, and another to record the front view of the car (Fig.1 a). Vehicular data such as GPS position and driving speed were collected using the Race Technology DL1 CLUB Data Logger (Fig.1 b), which was placed under the driver's seat. Physiological data, such as heart rate (HR), EDA, and temperature (TEMP), were collected through the Empatica E4 wristband (Fig.1 c), a user-friendly watch with built-in sensor capabilities. The E4 wristband was worn by the participant during the fixed course drive. The

sampling rates of HR, EDA, and TEMP are 1 Hertz, 4 Hertz, and 4 Hertz respectively.

### D. Physiological Signals

Physiological signals, such as heart rate and electrodermal activity, provide information regarding an individual's response to various environmental situations depending on how they interpret a given scenario. These signals can be measured using multiple methods, including body mounted sensors and wearable devices. For our study, we chose to measure and analyze signals that would provide the most information regarding physiological arousal in response to key driving and road events: HR, EDA, and TEMP.

Physiological arousal is an essential component of the body's response to a broad array of stimuli, such as external events and internal psychological processes. Physiological arousal may be due to a variety of factors including heightened attention, panic, anxiety, and stress, to name a few. Upon encountering a stimulus, an individual's body goes through a series of events while the situation is evaluated and ultimately provides a response (e.g., a boost of energy to equip the individual to face the forthcoming challenge). The response begins with the release of hormones such as adrenaline, cortisol, and noradrenaline, which elevates the level of physiological arousal. Heightened physiological arousal triggers a series of physiological changes including elevated heart rate, elevated blood pressure, sweat production, and a rise in body temperature to name a few [19, 24]. Each person's physiological arousal level to the same situation may differ based on their gender, age, temperament, prior life experiences, existing illness, and many more factors [20].

Physiological signals can provide information regarding a person's degree of arousal and response to various driving situations depending on how the driver may interpret a given scenario. These responses could include heightened attention to surroundings or a stress response to a challenging situation. Studying the physiological changes due to driving-related physiological arousal among seniors with and without elevated $A\beta$ may have the potential to reveal unique characteristics of those at risk of developing cognitive decline. Routine drives pose drivers with multiple scenarios that cause stress and many other emotions even without people realizing it. Situations such as heavy traffic loads, highways, lane changing, and road construction are all examples of driving stressors.

For our research, we chose to study HR, EDA, and skin TEMP, all of which are key identifiers of a person's level of physiological arousal that could occur during either signs of heightened attention or stress [24]. HR is defined as the number of times an individual's heart beats in sixty seconds. It is known that HR increases during physiological arousal, therefore, monitoring the HR of $A\beta$ positive and negative seniors while driving and comparing their responses will give us insights into unique variations that exist among each group. In addition, EDA has been broadly used for many years as an attested noninvasive identifier of stress [22]. As previously stated, sweating is also one of the key responses during physio-

logical arousal, which causes the skin's electrical conductivity, known as skin conductance, to change based on the amount of sweat secreted [23]. EDA measures skin conductance and provides valuable information on a person's level of potential stress. EDA consists of two different components: tonic and phasic. The tonic component, otherwise known as skin conductance level, is composed of the underlying characteristics and slowly fluctuating activity of the signal [23]. The phasic component, known as the skin conductance response, is the rapidly changing, short term segment that occurs within a span of a few seconds [23, 25]. Studying EDA and its components will provide information on the differences in the driving-related physiological arousal between $A\beta$ positive and negative seniors. Lastly, an increase in body temperature can also be attributed to arousal.

These three physiological signals can be easily and accurately measured using wearable technology. For our study, we used the user-friendly Empatica E4 wristband, which is known to provide precise and reliable digital biomarker data. In this study, our analysis targets HR, EDA, and TEMP responses in senior drivers, both with and without elevated $A\beta$. This focus may enable the identification of new risk biomarkers that are accessible, non-invasive, and cost-effective, potentially aiding the identification of those potentially at risk of cognitive decline.

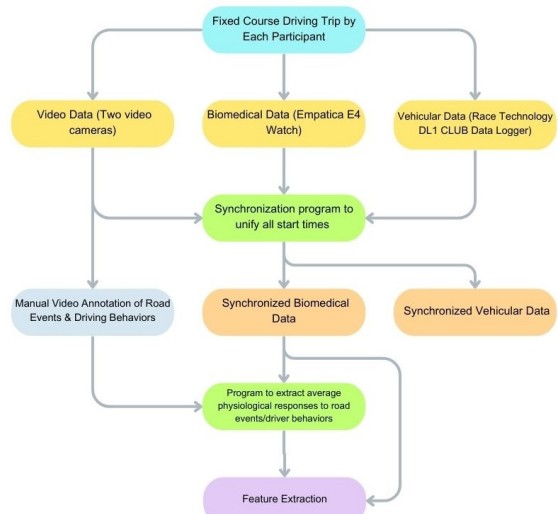

Fig. 2: Overview of Data Collection and Processing Procedure

### E. Data Processing

The road- and driver-view videos recorded during the fixed course driving trip of each participant were manually annotated to mark key road events along with their respective time windows, specifying the start and end times of each event. These windows are determined by the researcher who performs the manual annotation. Table I presents the full list of annotated road events and driver behaviors.

As not all events, such as hitting the curb, were experienced by each participant, only events that occurred in at least 70% of participants in each group were used in this study (i.e., a minimum of 20 participants in each group must have data present). Therefore, the road events and driver behaviors highlighted in bold in Table I were selected for analysis.

The average HR, EDA, and TEMP data were extracted for each road event highlighted in bold from Table I based on time windows that specify the start and end times of each driving event (e.g., average HR response during freeway entry ramp). This was done by first synchronizing the start times of the video and E4 wristband data using a Python script we developed. Then, to unify the sampling rates of all three physiological signals to 4 Hertz, we upsampled HR by performing linear interpolation. Lastly, to calculate the average physiological response to each road event and driver behavior, we developed a Python script to aggregate all occurrences of each driving behavior and road event and average their corresponding physiological signals (HR, EDA, and TEMP individually) to produce one average response value for each physiological signal. For example, one result from these calculations would be as follows: participant X had average HR, EDA, and TEMP responses of 85 beats per minute, 2.5 microsiemens, and 35 degrees Celsius respectively to left turns (road event 15). The data processing procedure is illustrated in Fig. 2.

In addition to averaged physiological responses to key road events, we extracted additional features, such as variances and differences between physiological responses to key road

TABLE I: All Road Events and Driver Behaviors

| Event ID | Road Event |
|---|---|
| 1 | **Freeway entry ramp** |
| 2 | **Freeway exit ramp** |
| 3 | Missed freeway entry |
| 4 | Missed freeway exit |
| 5 | **Red light** |
| 6 | **Green light** |
| 7 | Yellow light |
| 8 | Stop sign - driver did not stop |
| 9 | **Stop sign - driver stopped** |
| 10 | **Intersection with yield sign** |
| 11 | **Intersection without traffic signs** |
| 12 | **Light traffic load** |
| 13 | **Medium traffic load** |
| 14 | Heavy traffic load |
| 15 | **Left turn** |
| 16 | **Right turn** |
| 17 | **Lane change** |
| 18 | **Going straight** |
| 19 | **Roundabout** |
| 20 | Missed opportunity |
| 21 | Unsafe gap |
| 22 | Failure to yield |
| 23 | Out of lane |
| 24 | Run through red light |
| 25 | Run through yellow light |
| 26 | **Roll through stop sign** |
| 27 | Hit curb |
| 28 | Overtake |
| 29 | Pedestrian crossing the street |
| 30 | Pedestrian on the side of the road |
| 31 | Road construction |
| 32 | Construction signs |
| 33 | Multilane road |

events and benchmark events. We defined the following two benchmark events:

1) Average physiological response throughout the entire fixed course drive
2) Average physiological response over the 5 minutes preceding the fixed course trip, which is referred to as the baseline measure

We converted "roll through stop sign" (event 26) into a categorical variable with two options: "yes" (participant rolled through the stop sign without coming to a full stop) and "no" (participant did not roll through the stop sign and made a complete stop). The full list of 131 physiological features we extracted and used in all EoDT algorithms can be found in Table II.

57% of the participants had complete data for all 131 features while 43% had data missing for less than five features. Missing data was handled using the mean imputation method. For example, if HR data from a specific segment of the freeway is missing for a participant from one of the two study groups (e.g. due to a faulty sensor), then the missing data is imputed based on the average for that event from all participants in that individual's group. We used the same procedure for any events with missing data.

## IV. METHOD

### A. Ensembles of Decision Trees

EoDT are a popularly used ensemble learning method that amalgamate the predictions of multiple trees to provide a more accurate and reliable result than a single decision tree [17]. The ensemble learning technique in ML has demonstrated its ability to create more robust and accurate models that are less susceptible to overfitting. The three popular ensemble learning methods are bagging, boosting, and stacking [18]. For our study, we use multiple bagging and boosting techniques.

*Bagging (bootstrap aggregating):* This method uses bootstrapping to create multiple random subsets of the data and trains each model in parallel on one subset [18]. The final prediction is generated by taking a majority vote (for classification) or average (for regression) of the results generated by all the models. Bagging is known for its efficiency at reducing variance and overfitting.

*Boosting:* In this method, training of the models is performed in a sequential fashion, where every successive model learns from the errors of the prior one. Weights are assigned to the output of each tree, with higher weights given to incorrect predictions. This way, multiple weak learners are combined to generate one strong learner capable of providing results with higher accuracies [18, 26].

### B. Models Used

We use the following four EoDT models for potential risk biomarker discovery:

*1) Random Forest (RF):* This is a bagging algorithm which combines multiple decision trees that are fit on random subsamples of the training data. Each tree makes one prediction, and the final result is generated based on whether the task is classification or regression. A majority vote is used to determine the final class prediction for a classification task, and the results of all the trees are averaged to generate a final result for regression. Random Forests have shown high capability of reducing overfitting and provide high accuracy since the forecasts of all trees are combined to make a decision. However, since there are multiple trees and each tree needs to be trained separately, Random Forests require quite a bit of time and memory.

*2) Extremely Randomized Trees (ET):* Also known as Extra Trees, this algorithm is quite similar to Random Forest. Extra Trees also builds many decision trees, however, contrary to Random Forest, this algorithm trains each tree on the entire training data. Therefore, this algorithm does not fall into the bagging or boosting category. A subset of features is randomly selected for each tree and the node split is done based on a randomly selected threshold value for every feature. Since the splitting of trees is done in a random manner, this algorithm reduces variance and has faster training times than Random Forest.

*3) AdaBoost (AB):* Adaboost is a boosting algorithm which sequentially trains multiple weak classifiers and assigns higher weights to errors made by the preceding model in order for the following models to focus on misclassified samples. The algorithm starts by assigning the equal weights to all data samples. A weak classifier is then trained on this data and the weights of this classifier are calculated based on the errors produced. Next, each sample's weight is updated, where lower weights are assigned to accurately classified samples and higher weights are assigned to incorrectly classified samples. Weights are also assigned to the model where higher weight is given to classifiers with higher accuracies. Training, classifier weight calculation, and weights assignment are repeated until the stopping criterion is met. The final result for a classification task is calculated by taking a majority vote of all the weak learners. AdaBoost is quite a flexible model and can work with a range of base classifiers. It also reduces the risk of overfitting due to its nature of assigning higher weights to misclassified samples. On the other hand, AdaBoost is a bit time intensive and is sensitive to data which is noisy or contains outliers.

*4) XGBoost (XGB):* XGBoost stands for Extreme Gradient Boosting and is a boosting algorithm widely used for classification and regression. XGBoost is similar to AdaBoost, but uses only decision trees as its base learner. In addition, the weights of each sample are calculated using gradient descent in XGBoost. Unlike the other models, XGBoost can handle missing data and finds the optimal method of imputing the missing data sample. Additionally, the algorithm is well known for its speed and reduction of the overfitting problem due to its use of L1 and L2 regularization techniques. Conversely, XGBoost may be subject to overfitting when working with small datasets. Also, precise hyperparameter tuning is crucial to attain the best performance using XGBoost.

TABLE II: Input Features for All Four EoDT Algorithms

| Feature Name | Description | Number of Features |
|---|---|---|
| HR_road event ID number (e.g. "HR_1") | Average HR response to road events 1,2,5,6,9,10,11,12,13,15,16,17,18,19 | 14 |
| EDA_road event ID number (e.g. "EDA_1") | Average EDA response to road events 1,2,5,6,9,10,11,12,13,15,16,17,18,19 | 14 |
| TEMP_road event ID number (e.g. "TEMP_1") | Average TEMP response to road events 1,2,5,6,9,10,11,12,13,15,16,17,18,19 | 14 |
| HR_var | Variance in HR throughout the fixed course drive | 1 |
| EDA_var | Variance in EDA throughout the fixed course drive | 1 |
| TEMP_var | Variance in TEMP throughout the fixed course drive | 1 |
| diff_HR_road event ID number (e.g. "diff_HR_1") | Difference between average HR throughout the drive and average HR response to road events 1,2,5,6,9,10,11,12,13,15,16,17,18,19 | 14 |
| diff_EDA_road event ID number (e.g. "diff_EDA_1") | Difference between average EDA throughout the drive and average EDA response to road events 1,2,5,6,9,10,11,12,13,15,16,17,18,19 | 14 |
| diff_TEMP_road event ID number (e.g. "diff_TEMP_1") | Difference between average TEMP throughout the drive and average TEMP response to road events 1,2,5,6,9,10,11,12,13,15,16,17,18,19 | 14 |
| basediff_HR_road event ID number (e.g. "basediff_HR_1") | Difference between baseline HR and average HR response to road events 1,2,5,6,9,10,11,12,13,15,16,17,18,19 | 14 |
| basediff_EDA_road event ID number (e.g. "basediff_EDA_1") | Difference between baseline EDA and average EDA response to road events 1,2,5,6,9,10,11,12,13,15,16,17,18,19 | 14 |
| basediff_TEMP_road event ID number (e.g. "basediff_TEMP_1") | Difference between baseline TEMP and average TEMP response to road events 1,2,5,6,9,10,11,12,13,15,16,17,18,19 | 14 |
| roll_stop | Participant rolled through the stop sign (event 26) without coming to a full stop (yes/no) | 2 |

TABLE III: Performances of All Four EoDT Models

| EoDT Models | Accuracy (%) | | Precision | | Recall | | F1 Score | |
|---|---|---|---|---|---|---|---|---|
| | Train | Test | Train | Test | Train | Test | Train | Test |
| Random Forest | 100 | 83.33 | 1.0 | 0.838 | 1.0 | 0.831 | 1.0 | 0.832 |
| Extra Trees | 100 | 77.78 | 1.0 | 0.784 | 1.0 | 0.775 | 1.0 | 0.775 |
| AdaBoost | 100 | 77.78 | 1.0 | 0.777 | 1.0 | 0.777 | 1.0 | 0.777 |
| XGBoost | 100 | 79.63 | 1.0 | 0.797 | 1.0 | 0.795 | 1.0 | 0.796 |

*C. System Training, Evaluation, and Extraction of Significant Features*

For each of the four EoDT models, denoted as $M_i$ ($i = 1, 2, 3, 4$), we applied K-fold cross-validation (CV) with K=5 to the data and features described in Table II. During K-fold CV, we partitioned the dataset into K partitions using stratified random sampling to ensure proportional representation of all classes in each fold, maintaining balance throughout training and testing. The training and testing process takes K iterations and every iteration has its unique training and testing set. At each fold, the system is trained using the training set, which contains K-1 partitions, and evaluated using the testing set, which takes the remaining partition. For 5-fold CV, at each iteration, four partitions are used for training and the remaining partition is used for testing. The K-fold CV method provides a robust estimate of the model's performance on unseen data. In each fold, the training set contained approximately 21 $A\beta$ positive and 22 $A\beta$ negative samples, while the testing set contained approximately five $A\beta$ positive and six $A\beta$ negative samples. To evaluate the performance of all models during training, we used the following evaluation metrics: accuracy, precision, recall, and F1 score. The same evaluation metrics along with confusion matrices were computed at each fold on the partition used as the test data, i.e. the partition that is not used as a part of training at that fold. The evaluation results across all K folds were averaged to produce the overall performance results generated by K-fold CV.

After the training phase of each EoDT model, we generated a ranking of feature importance at each fold. The ML algorithm used in each model calculates feature importance across the entire model by aggregating the contribution of each feature to the predictive performance of the trees. The four EoDT models used in this research employed different methods to calculate feature importance — Random Forest used entropy, AdaBoost used Gini impurity, XGBoost used gain, and Extra Trees also used Gini impurity. In these model learning processes, each feature is assigned a numerical value corresponding to its significance within the model.

The process of extracting significant features is described as follows. For each of the four EoDT models $M_i$, at every fold K (the index of the folds in 5-fold CV processes, where $K = 1, 2, 3, 4, 5$), we extract the top 25% of the features, denoted as $M_{i,F_K}$. Here, $F_K$ denotes the top 25% of features extracted at fold K. The top ranked features from each model $M_i$ are found by taking the intersection of the top ranked features generated at each fold, i.e., $M_{i,F} = M_{i,F_1} \cap M_{i,F_2} \cap M_{i,F_3} \cap M_{i,F_4} \cap M_{i,F_5}$. This step generated four separate feature lists, one for each model ($M_{1,F}, M_{2,F}, M_{3,F}, M_{4,F}$). Lastly, the features that appeared in the feature lists of three or more models were chosen as the risk biomarkers most effective at identifying senior drivers with $A\beta$ positivity: Risk biomarkers = $(M_{1,F} \cap M_{2,F} \cap M_{3,F})$ or $(M_{1,F} \cap M_{2,F} \cap M_{4,F})$ or $(M_{1,F} \cap M_{3,F} \cap M_{4,F})$ or $(M_{2,F} \cap M_{3,F} \cap M_{4,F})$ or $(M_{1,F} \cap M_{2,F} \cap M_{3,F} \cap M_{4,F})$.

## V. EXPERIMENTAL RESULTS

The accuracy, precision, recall, and F1 score of all four models were evaluated in our experiments, and the results are shown in Table III and illustrated in Fig. 3 (the standard deviation across all five folds for each metric is shown as error bars). The Random Forest algorithm demonstrated superior performance compared to the other three models, achieving an

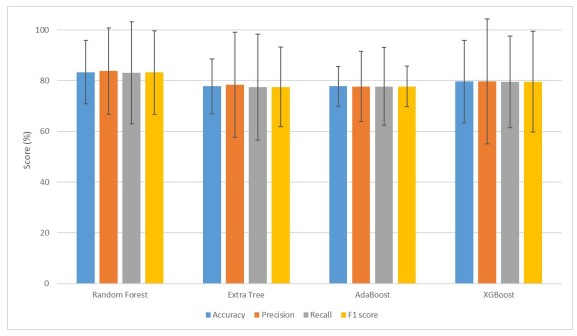

Fig. 3: Accuracy, Precision, Recall, and F1 Score of All Models

accuracy of 83.33%, along with the highest scores in precision, recall, and F1 metrics. All models achieved a full score on all evaluation metrics during training. XGBoost ranked second in performance following Random Forest. AdaBoost and Extra Trees were the lowest performers, producing scores that were very similar to each other across all four evaluation metrics. Extra Trees had a slightly higher precision than AdaBoost while AdaBoost had a marginally higher recall and F1 score. When comparing the performances of both boosting models, XGBoost and AdaBoost, both were relatively similar, with XGBoost showing comparatively higher results across all evaluation metrics. The confusion matrices of all models are shown in Fig. 5.

Using the aforementioned method, we discovered five potential novel susceptibility biomarkers which identified individuals with elevated $A\beta$ burden as compared to those without. The discovered biomarkers are presented in Table IV and the importance of each top-ranked feature generated by each of the four models is shown in Fig. 4.

Studying these discovered potential biomarkers in depth revealed some characteristics unique to $A\beta$ positive seniors compared to their $A\beta$ negative peers. HR appears to be the most effective physiological indicator for distinguishing older drivers with elevated $A\beta$ from those without, outperforming responses from EDA and TEMP, with TEMP being the least effective. Intersections without traffic signs emerge as a pivotal road event that triggers noticeable differences in HR responses between $A\beta$ positive and negative seniors. Seniors with $A\beta$ positivity exhibit a higher average HR during this road event than their

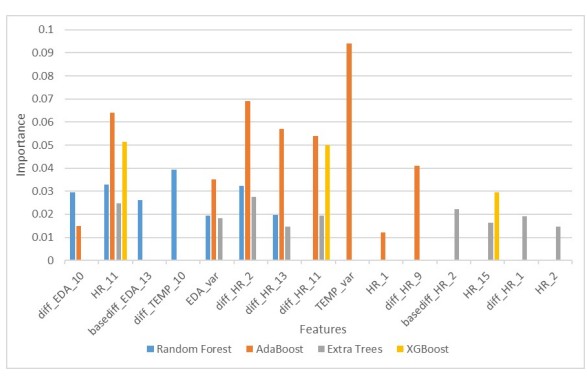

Fig. 4: Importance of Each Top-Ranked Feature

TABLE IV: Novel Risk Biomarkers Discovered in this Study

| Feature Name | Description | Selected by Models |
|---|---|---|
| HR_11 | Average HR while encountering intersections without traffic signs | XGB, RF, ET, AB |
| diff_HR_11 | Difference between average HR throughout the drive and average HR response while encountering intersections without traffic signs | XGB, ET, AB |
| diff_HR_2 | Difference between average HR throughout the drive and average HR response to freeway exit ramps | RF, ET, AB |
| diff_HR_13 | Difference between average HR throughout the drive and average HR response to medium traffic loads | RF, ET, AB |
| EDA_var | Variance in EDA throughout the fixed course drive | RF, ET, AB |

$A\beta$ negative peers. Furthermore, while encountering intersections without traffic signs, freeway exit ramps, and medium traffic loads, the average HR of seniors with elevated $A\beta$ differs much less than their average HR throughout the fixed course drive when compared to seniors with no elevation in $A\beta$. In other words, $A\beta$ negative seniors seem to experience a greater change in HR in the same situation than $A\beta$ positive seniors. Lastly, senior drivers with elevated $A\beta$ have a higher variance in their EDA responses throughout the fixed course drive, which could be indicative of more pronounced fluctuations in arousal levels when compared to $A\beta$ negative seniors. Overall, the physiological arousal levels of those with elevated $A\beta$ are quite different during intersections without traffic signs, freeway exit ramps, and medium traffic loads when compared with those with no elevation in $A\beta$. In these situations, $A\beta$ positive seniors show measurable differences in HR and EDA, with HR being the most significantly affected parameter. Measuring senior drivers' physiological responses to these road events may provide insights into their risk of developing cognitive decline attributed to AD in the forthcoming years. Such individuals can work with their healthcare providers to arrange for regular monitoring to be vigilant for any early signs of cognitive decline.

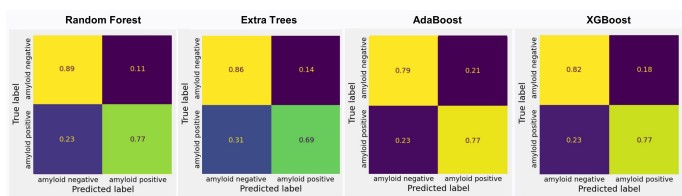

Fig. 5: Comparison of Confusion Matrices for All Four Ensembles of Decision Trees Models

## VI. CONCLUSION

Early diagnosis of AD is crucial to begin the necessary treatment plans in order to delay progression of the disease. There is a great need for more cost-efficient, non-invasive, and reliable methods to identify individuals at greater risk of developing AD in the future. Testing for the presence of biomarkers has shown a promising path to early risk detection, however, current AD biomarker screening involves invasive and expensive procedures, making it inaccessible to some groups. Risk biomarkers that can be non-invasively and inexpensively measured using machine learning technologies will allow individuals at higher risk to be identified.

In this study, we proposed a machine learning approach for early detection of those at risk of cognitive decline by studying

senior drivers' physiological responses to cognitively complex driving events. The focus was transferred from brain-derived biomarkers to physiological biomarkers, to investigate the differences in physiological arousal among $A\beta$ positive and negative senior drivers. We discovered five potential risk biomarkers of cognitive decline associated with $A\beta$ positivity by analyzing physiological responses to driving using Ensembles of Decision Trees models. We employed one bagging model, Random Forest, two boosting models, AdaBoost and XGBoost, and the Extra Trees algorithm to perform the classification task of identifying $A\beta$ positive seniors. Accuracy, precision, recall, and F1 score were calculated for all models during training and testing using 5-fold cross validation. The highest accuracy of 83.33% was achieved by Random Forest. XGBoost was the next best performing model with an accuracy of 79.63%, while Extra Trees and AdaBoost were the lowest performing algorithms with accuracies of 77.78%. Novel risk biomarkers were identified by systematically combining the top ranked features over all five folds for each model and then selecting the features appearing in the list of top-ranked features of three or more models. The discovered risk biomarkers demonstrate that seniors with elevated $A\beta$ have observable differences in HR and EDA responses to the following road events: intersections without traffic signs, freeway exit ramps, and medium traffic loads. Following our approach, studying physiological signals during the cognitively demanding task of driving shows a promising path for the discovery of potential risk biomarkers of AD that can be non-invasively and cost-efficiently measured. Such biomarkers can be used to recognize seniors at greater risk of developing MCI due to AD in the forthcoming years and arrangements can then be made for more invasive testing if needed (e.g., PET, CSF).

We are actively recruiting more $A\beta$ positive and negative participants along with young participants and those with MCI. Additionally, we are conducting a two-year follow-up study for each participant. In the near future, we plan to validate and improve the generalizability of our findings by using data from this growing and more diverse group. As next steps, we also aim to include additional physiological signals and road events, enhance model performance, and study additional ML models to discover more risk biomarkers of cognitive decline.

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
