# OpenReview forum: "Investigating Physiological Responses During Driving as Potential Biomarkers of Cognitive Decline in Seniors Using Decision Tree Ensemble Modeling"
_IEEE.org/EMBS/BHI/2024/Conference — IEEE BHI'24_

### Official Review · Reviewer_RiSf · 2024-08-08
**The paper is well-written with detailed data collection, clear motivation and interesting insights. The dataset, if publicly released, would benefit the research community. Improvements are needed in the related work section**

**Overall Rating:** 8
**Confidence:** 5

**Other Quality Metrics:**

(a) Clarity of writing = excellent
 (b) Clinical Significance = good
(c) Methodological Novelty = good
(d) Experiments and Results = great

**Questions For The Authors:**

Major revision:

1-	The related work section does not cite the most relevant literature. For instance, works that extract genetic biomarkers using decision trees are not the most relevant. It would be nice to cite literature that extracts physiological signals for Alzheimer’s or other similar diseases. Additionally, there are multiple works that use other machine learning models, like CNN, etc., for extracting physiological features or biomarker discovery for Alzheimer’s specifically. Please cite these works and highlight the contribution of the paper compared to these works.

2-	At the end of section III-E, where # of features (131) are mentioned, please also include how many total samples were available for training and testing the model. At the end of the data processing pipeline, is the dataset still balanced?

Minor Issues/Recommendations to improve the paper:

1-  As not all events, such as hitting the curb, were experienced by each participant, only events that occurred in at least 70% of participants in each group were used in this study
(i.e., a minimum of 20 participants in each group must have data present).”
It would be interesting to see how much data/features were available for each pt. and how that missingness was dealt with.

2-	Figure 3:  The line plot is appropriate when there is a relationship between values on x-axis. I recommend presenting these results using a bar plot. Please also report the standard deviation across folds for each metric either in Table III or as error bars on the bar plot.

3-	Briefly mention how were hyperparameters (depth of tree, # of trees etc.) tuned? Usually, this procedure is done through K-fold validation first (only training + validation data), and then the best model is trained and tested with multiple random splits of the data, and average and std performance is reported. However, in this paper, K-folds is used to test models’ generalization to unseen data.

4-	“For each of the four EoDT models Mi, at every fold K, we extract the top 25% of the features, denoted as Mi,FK (K = 1, 2, 3, 4, 5).”
What is Fk and K?

5-	“Lastly, the features that appeared in the feature lists of three or more models were chosen as the risk biomarkers most effective at identifying senior drivers with Aβ positivity:”
Why were intersections in 3 models chosen and not in all 4 models? Since some models perform better than others, consider using a weighted scheme to put more weight on features provided by the best model. Given the already substantial length of the paper, this could be addressed in future work.

**Strengths:**

1-	The paper is very well-written. Data collection procedures are described in sufficient detail. The flow of paper is smooth. The motivation and problem statement are clearly articulated.

2-	A multimodal dataset is collected from a senior population in a rigorous manner, ensuring fairness and balance in distribution. Publicly releasing this dataset would significantly benefit the research community.

3-	Data Processing and Machine Learning modeling are explained in detail. The paper considers both average and std in the physiological signals.

4-	The reported important features provide interesting insights and enhance the interpretation of the proposed method.

**Summary Of The Paper:**

This paper explores a machine learning approach to identify non-invasive, cost-effective biomarkers for Alzheimer's disease risk by analyzing physiological responses during driving tasks. Researchers collected data using empatica watch, a video camera, and a data logger measuring heart rate, electrodermal activity, temperature, and driving behavior in seniors with and without elevated PET beta-amyloid levels. They classified data from 54 participants using decision tree ensemble techniques and identified five significant biomarkers. This method offers the potential for early Alzheimer's detection by leveraging driving-related physiological responses.

**Weaknesses:**

1-	The related work can be significantly improved. I recommend focusing on works that extract physiological features and methods specifically aimed at Alzheimer’s biomarker discovery.

2-	The paper places considerable emphasis on ensemble models. However, other models like shallow MLP and SVM could also serve as potential benchmarks and could be included. Given the already substantial length of the paper, this could be addressed in future work.


3-	There are some concerns with dataset suffering from the curse of dimensionality. The feature space (131) is larger than the sample space (I assume it’s less than 54?). If that’s the case, some feature selection should be done before data is fed to the ML models.

---

### Official Review · Reviewer_hrr6 · 2024-08-14
**Investigating Physiological Responses During Driving as Potential Biomarkers of Cognitive Decline in Seniors Using Decision Tree Ensemble Modeling**

**Overall Rating:** 8
**Confidence:** 4

**Other Quality Metrics:**

(a) excellent
(b) excellent
(c) great
(d) great

**Questions For The Authors:**

Could you please explain how did you deal with data missingness? How did you decide the time window size to calculate the average HR? Is it one minute before the event or 30 seconds before the event and 30 seconds after the event?

**Strengths:**

Although the beta-amyloid is not the gold standard for the diagnosis but only an indicator, the combination of driving and wearables can be an attempt to prevent AD.

The data collection and processing procedure is solid with steps like upsampling and time synchronization.

**Summary Of The Paper:**

This paper presents a machine learning approach to identify potential risk biomarkers for Alzheimer’s disease (AD) by analyzing physiological responses during driving tasks. The authors focused on cognitively normal seniors, comparing those with elevated beta-amyloid (Aβ) levels (a marker associated with AD risk) to those without elevated levels. Physiological data, including heart rate, electrodermal activity, and temperature, were collected during driving events (like right turns and roundabouts) to explore differences between the two groups. Five risk biomarkers are found to differentiate the two groups of seniors.

**Weaknesses:**

(a) The visualization of the features could be improved by plotting a bar plot that shows the significance of each feature, which will increase the interpretability of models.
(b) The data is recorded with a determined route instead of a random route from the participant's daily life. The conclusion of this paper has not been tested during daily life.

---

### Official Review · Reviewer_zQQ2 · 2024-08-29
**Investigating Physiological Responses During Driving as Potential Biomarkers of Cognitive Decline in Seniors Using Decision Tree Ensemble Modeling.  This paper presents an innovative method for identifying potential Alzheimer's disease risk biomarkers using physiological responses during driving. While the approach is promising and the methodology is sound, the small sample size and lack of longitudinal validation limit its immediate applicability. The study provides a foundation for future research in early AD detection, but requires further investigation with larger cohorts.**

**Overall Rating:** 7
**Confidence:** 4

**Other Quality Metrics:**

(a) Clarity of writing: Good
The paper is well-structured and generally clear, but some technical aspects could be explained more thoroughly for a broader audience.

(b) Clinical Significance: Good
If validated, this approach could lead to more accessible early screening for AD risk. However, more research is needed to confirm its clinical utility.

(c) Methodological Novelty: Excellent
The use of driving-related physiological responses as potential AD risk biomarkers is highly innovative and opens up new directions for research.

(d) Experiments and Results: Good
The experimental design is sound, and the results are promising. However, the small sample size and lack of longitudinal data limit the strength of the conclusions.

**Questions For The Authors:**

1. Have you considered conducting a follow-up study to track these participants over time and see if the identified physiological responses actually predict future cognitive decline?

2. How do you plan to validate these findings with a larger, more diverse group of participants?

3. What challenges do you foresee in translating this research into a practical screening tool that could be used in everyday clinical settings?

**Strengths:**

1. Novelty

The study presents an innovative approach to identifying potential biomarkers for Alzheimer's disease risk by analyzing physiological responses during driving, a complex cognitive task.

2. Methodology

The use of multiple ensemble decision tree models (Random Forest, Extra Trees, AdaBoost, and XGBoost) for classification and feature importance ranking is a robust approach. The 5-fold cross-validation adds credibility to the results.

3. Standardized Data Collection

The study uses a comprehensive set of data (video, vehicular, and physiological) collected during a standardized driving course, which helps control for variables and increases reliability.

4. Non-invasive Approach

The proposed method offers a potentially cost-effective and non-invasive way to screen for AD risk, which could have significant clinical implications if validated.

5. Clear Presentation

The paper is well-structured, with clear explanations of the methodology, results, and implications. Tables and figures effectively supplement the text.

6. Practical Implications

The identified biomarkers, if validated, could provide a more accessible method for early AD risk screening, potentially leading to earlier interventions.

**Summary Of The Paper:**

This study investigates the use of physiological responses during driving as potential biomarkers for identifying seniors at risk of cognitive decline due to Alzheimer's disease (AD). The research compares the physiological responses of cognitively normal seniors with and without elevated beta-amyloid (Aβ) levels, as detected by PET scans.

The study involved 54 participants (26 Aβ positive, 28 Aβ negative) who completed a standardized 7.1-mile driving course. Physiological data (heart rate, electrodermal activity, and temperature) were collected using wearable technology. The researchers applied four ensemble decision tree models (Random Forest, Extra Trees, AdaBoost, and XGBoost) to analyze the data and identify potential biomarkers.

The analysis yielded five potential risk biomarkers, primarily related to heart rate responses during specific driving events such as intersections without traffic signs, freeway exit ramps, and medium traffic loads. The Random Forest model achieved the highest classification accuracy of 83.33%.

The authors propose that these physiological responses during driving could serve as non-invasive, cost-effective biomarkers for identifying individuals at higher risk of developing AD. They acknowledge the need for further validation and longitudinal studies to confirm the predictive value of these biomarkers.

The paper concludes by suggesting that this approach could potentially lead to more accessible screening methods for AD risk, potentially allowing for earlier intervention and better management of disease progression.

**Weaknesses:**

1. Small Sample Size

The study uses a relatively small cohort (54 participants), which limits the statistical power and generalizability of the findings.

2. Lack of External Validation

The identified biomarkers have not been validated on an independent dataset, which is crucial for confirming their reliability and generalizability.

3. Absence of Longitudinal Data

Without follow-up data, it is unclear whether the identified biomarkers actually predict future cognitive decline.

4. Potential Confounding Factors

The wide age range (65-85 years) and gender imbalance between groups could introduce confounding variables that are not fully addressed.

5. Model Performance

While promising, the 83.33% accuracy leaves room for improvement, and there is limited discussion on potential ways to enhance model performance.

---

### Decision · Program_Chairs · 2024-09-23

Accept